# BushenHuoxue formula promotes osteogenic differentiation via affecting Hedgehog signaling pathway in bone marrow stem cells to improve osteoporosis symptoms

Yuqi Chen[1☯], ZhiYong Wei[2☯], HongXia Shi[3], Xin Wen[4], YiRan Wang[1], Rong Wei [1,3]*

1 Department of the People's Hospital of Suzhou New District, Suzhou, China, 2 Kuitun Hospital of Xinjiang Production and Construction Corps, Xinjiang Uygur Autonomous Region, China, 3 The Fourth Affiliated Hospital, Xinjiang Medical University, Urumqi, China, 4 Urumqi Friendship Hospital, Urumqi, PR China

☯ These authors contributed equally to this work.
* 952257684@qq.com

**Data Availability Statement:** All relevant data are within the paper and its Supporting Information files.

## Abstract

### Background

The BushenHuoxue formula (BSHX) has been previously demonstrated to ameliorate osteoporosis, but the mechanisms underlying this phenomenon are currently unclear. The present study aims at investigating the mechanisms that BSHX induces osteogenesis.

### Methods

We established an osteoporosis model in rats by bilateral ovariectomy and then treated the rats with an osteogenic inducer (dexamethasone, β-sodium glycerophosphate and Vitamin C) and BSHX. After that, bone marrow density and histopathological bone examination were evaluated by using HE staining and immunohistochemistry, respectively. We also assessed the differentiation of bone marrow mesenchymal stem cells (BMSCs) into osteoblasts by using immunofluorescence staining. ALP, BMP, and COL1A1 levels were determined by ELISA. We identified genes involved in pathogenesis of osteoporosis through Gene Expression Omnibus (GEO) database and subsequently selected Hedgehog signaling-related genes *Shh*, *Ihh*, *Gli2*, and *Runx2* for assessment via qRT-PCR and ELISA, Western blotting. Network pharmacology analysis was performed to identify bioactive metabolites of BSHX.

### Results

BSHX treatment in osteoporosis model rats promoted tightening of the morphological structure of the trabecular bone and increased the bone mineral density (BMD). BSHX also increased levels of osteoblast makers ALP, BMP, and COL1A1. Additionally, bioinformatics analysis of the GEO dataset showed that Hedgehog signaling pathway was involved in pathogenesis of osteoporosis, especially related genes *Shh*, *Ihh*, *Gli2*, and *Runx2*. Remarkably, BHSX upregulated these genes indispensably involved in the osteogenesis-related

**Funding:** This work was supported by National Natural Science Foundation grant (No. 81460676) and Medical and Health science and Technology project of Suzhou High-tech Zone(No. 2019Z011) and the 55th batch of funding for the China Postdoctoral Science Foundation Project.

**Competing interests:** The authors have declared that no competing interests exist.

**Abbreviations:** ALP, alkaline phosphatase; BMD, bone mineral density; BMP, Bone Morphogenetic Protein; BMSCs, bone marrow mesenchymal stem cells; BSHX, BushenHuoxue;Col1a1, type I collagen; DEGs, differentially expressed genes; ELISA, Enzyme-linked immunosorbent assay; FBS, fetal bovine serum; GEO, gene expression omnibus; GLI2, oncogene homolog 2; HE, hematoxylin-eosin; IHH, indian hedgehog; OP, osteoporosis; PBS, Phosphate-buffered saline; qRT-PCR, Quantitative real-time polymerase chain reaction; RUNX2, runt-related transcription factor 2; SD, sprague Dawley; SHH, sonic hedgehog; SPF, specific pathogen free; TCM, traditional Chinese medicine.

Hedgehog signaling pathway in both bone tissue and BMSCs. Importantly, we identified that quercetin was the active compounds that involved in the mechanism of BSHX-improved OP via affecting Hedgehog-related genes.

## Conclusion

Our results indicate that BSHX promotes osteogenesis by improving BMSC differentiation into osteoblasts via increased expression of Hedgehog signaling-related genes *Shh*, *Ihh*, *Gli2*, and *Runx2*, and quercetin was the bioactive compound of BSHX.

## 1. Introduction

Osteoporosis (OP) is caused by an imbalance of bone resorption and formation. It may profoundly affect patients' quality of life. Osteoblasts are the main functional cells involved in bone formation and metabolism. It plays an important role in the maintenance of the physiological function of the adult skeletal system [1]. The majority of traditional treatments for OP exert their effects via inhibition of osteoclasts, which are involved in degradation of bone [2, 3]. The therapeutic effects, however, is often unsatisfactory due to limited effectiveness and the presence adverse effects. It has been verified that the use of osteoclast inhibitory drugs may increase the risk of osteonecrosis and tumor occurrence [4]. Therefore, novel effective drugs to improve osteogenesis and treat OP are urgently required.

Traditional Chinese medicine (TCM) has a long history for the treatment of osteoporosis [5]. Studies have found out that certain traditional Chinese medicines can increase bone mass, area and density of the trabecular bone, and osteocalcin secretion, thereby ameliorating osteoporosis [6]. Bushenhuoxue (BSHX) formula is a traditional herbal decoction composed of nine herbs (listed in S1 Table). It has been widely used to treat various bone diseases in China for several decades [7]. According to TCM syndromes, the deficiency and blood stasis are regarded as the main pathogenesis of OA [8]. Of note, BSHX has been previously shown to dilate blood vessels and improve microcirculation, thereby modifying platelet aggregation, protecting vascular endothelial cells, and resisting oxidation [9]. CCK8 assay evaluation of the proliferation, differentiation, and drug toxicity of BSHX in bone marrow mesenchymal stem cells (BMSCs) revealed that 150 ug/mL BSHX were able to effectively elicit proliferation in BMSCs [6, 10]. In the previous study, we found that BSHX formula can attenuate bone loss and bone structure destruction in ovariectomized (OVX) mice. However, its pharmacological mechanism remains unclear, which limited the usage of BSHX in clinic. BMSCs are the main precursor of osteoblasts. A reduced differentiation of BMSCs into osteoblasts can lead to the reduction in bone formation, which is regarded as one of the etiological bases of OP [11–13]. It has previously been found that BMSC differentiation pathways in the bone marrow of OP patients are disrupted, resulting in reduced production of osteoblasts [14]. BMSC differentiation is regulated by multiple signaling pathways, including Wnt, BMP/Smad, Notch, and Hedgehog signaling pathways [15–17]. Hedgehog signaling plays an important role in embryonic development and tissue homeostasis by regulating cell growth. Importantly, the Hedgehog signaling pathway regulates differentiation of BMSCs and promotes osteoblasts differentiation while inhibiting adipocyte differentiation, thus promoting bone resorption and formation [18]. Furthermore, abnormalities in the Hedgehog signaling pathway may lead to the occurrence of OP [19]. Sonic hedgehog (SHH) and indian hedgehog (IHH) are important signaling molecules of the Hedgehog signaling pathway involved in regulating limb

development and osteoblast differentiation [20]. Both SHH and IHH can stimulate the differentiation of mesenchymal cell lines into osteoblasts, with SHH being more effective than IHH [21]. Additionally, the Hedgehog signaling pathway can also regulate the osteogenic differentiation of BMSCs by influencing key molecules of nuclear transcription factors glioma-associated oncogene homolog 2 (GLI2) and Runt-related transcription factor 2 (RUNX2) [22].

To investigate the pathogenesis of osteoporosis and the underlying mechanisms of BSHX in improving OP, we explored the role of BSHX prescription in OP at multi-aspects. Herein, we found that BSHX could promote the BMSC differentiate into osteoblasts and increase the bone mineral density of OP patients. Then, we identified that BSHX could trig the upregulation of IHH, GLI2 and RUNX2 of protein level in bone tissue of OP patients. Meanwhile, BSHX triggered SHH and GLI2 to upregulate the protein level in BMSCs, which could promote osteoblast differentiation, leading to the relief of OP. Of note, we performed network pharmacology analysis on BSHX, and found that quercetin was the active compounds that involved in the mechanism of BSHX-improved OP via affecting Hedgehog-related genes. Overall, the present study elucidates the molecular mechanisms of BHSX in OP treatment, providing insights into the potential therapeutic use of BSHX in OP.

## 2. Materials and methods

### 2.1 Animals

A total of 58 Specific pathogen Free (SPF) healthy female Sprague Dawley (SD) rats (6 weeks old, with a body mass of about 150 ± 20 g) were purchased from the Experimental Animal Center of Xinjiang Medical University [Animal License Number: SYXK (New) 2018–0003]. All animal experiments were carried out at the Animal Laboratory of Xinjiang Medical University according to international AAALAC standards and were approved by Experimental Animal and Use Management Committee of the First Affiliated Hospital of Xinjiang Medical University (Ethics approval number: IACUC20150210-07).

### 2.2 Sample collection

After intragastric administration of BHSX for 12 weeks, the rats fasted and were anesthetized with intraperitoneal injection of pentobarbital sodium. Fresh blood from the abdominal aorta was collected, centrifuged, and the upper serum was taken and frozen at -80˚ for use. After blood samples were collected, the rats were killed, the femur and the surrounding soft tissues of the rats were separated. The left femur was placed in a dual-energy X-ray absorptiometry to measure the bone mineral density (BMD). The right femur was placed in a 10% neutral formalin solution and fixed for 48 h for decalcification.

### 2.3 Preparation of BushenHuoxue (BSHX)

BSHX was prepared by the Traditional Chinese Medicine Hospital Affiliated to Xinjiang Medical University (S1 Table). Briefly, 10 kg of BSHX formula: antler (1.13kg), epimedium (1.13kg), psoralen (1.13kg), salvia (2.26kg), astragalus (2.26kg), radix rehmanniae (0.96kg), white peony root (0.96kg), and other Chinese medicines (0.17kg) were mixed with 1,500 mL boiling (100˚C) distilled water and incubated by shaking at room temperature for 1h. The mixture was centrifuged at 15,000 rpm for 10 min and the precipitate was discarded while the residue was extracted. The supernatant was filtered through a 0.22μm membrane, decompressed, and concentrated at 60˚C. The powder was stored at -20˚C. The extracted medicine contained 1.43 g/mL of raw active compound [51].

## 2.4 Establishment of the osteoporosis (OP) model and grouping

The SD rats were randomly divided into a sham operation group (blank group; n = 10) and an operation group (n = 48). Briefly, after anesthetization with a 1:1 mixture of Zoletil (50 mg/mL) and Lumianning (Xylazine Hydrochloride Injection, Hebei Vying Animal Pharmaceutical Co., Ltd) by intramuscular injection (0.02 mL/100 g body weight) followed by intraperitoneal injection of chloral hydrate solution (330 mg/kg body weight), ovaries of rats were bilaterally removed by surgery. After establishment of the model, eight animals died. The remaining rats were randomly divided into OP model group, BSHX group, estradiol group, and calcium group, with 10 rats in each group. Rats in the BSHX group were gavaged with 5 g BSHX/kg bodyweight, while rats in the blank (sham) group and OP model group received equal volumes of saline. In the estradiol and calcium groups, animals received 1 mg/kg bodyweight estradiol and 20 mg/kg bodyweight calcium, respectively. All compounds were administered intragastrically once daily for 12 consecutive weeks at 7 days after operation.

## 2.5 Isolation, culture, and identification of rat BMSCs

The 6-week-old female SD rats were sacrificed by cervical dislocation. The femur and tibia were removed under sterile conditions, and the epiphysis at both ends was cut with scissors. The bone marrow was flushed out with a syringe and placed on a sterile plate containing Phosphate-buffered saline (PBS) and 200 U/mL penicillin. After centrifuging at 1,000 rpm for 5 min, α-MEM medium containing 1% nafcillin and 10% fetal bovine serum (FBS, Gibco) was added to resuspend the cell pellet. The cells were inoculated into a culture flask, and the medium was changed for the first time after 24 h followed by media changes every 3 days. The third generation of cells was used for subsequent experiments, and flow cytometry was utilized to detect the levels of CD29, CD44, CD34, and CD45 for identification of rat BMSCs [23].

## 2.6 Induction of BMSC differentiation

BMSCs were split into a blank group, a BSHX group, an estradiol (E2) group and an osteogenic inducer group. The BSHX group was treated with BSHX at 150 ng/mL [6, 10]. The E2 group was treated with 17-β-estradiol (Sigma Chemical Co., St. Louis, MO, USA) at a concentration of $10^{-8}$ mol/L, and the osteogenic inducer group was treated with a combination of $1*10^{-8}$ mol/L dexamethasone, 10 mmol/L β-glycerophosphate sodium and 50 µg/mL ascorbic acid. The blank group was left untreated. The induction times were 1, 3, 5 and 7 days, with the medium being changed every 3 days.

## 2.7 Hematoxylin-Eosin(HE) staining

HE staining was used to observe the micromorphological structures of the femoral head in rats. Briefly, sections were dewaxed, dehydrated, stained with HE, transparentized, and mounted. An OLYMPUS BX51 microscope (OLYMPUS, Japan) was used to observe morphology at a magnification of 100x and acquire images.

## 2.8 Immunofluorescence

After culturing BMSCs for 7 days, cells were fixed, incubated with 0.5% Triton X-100 for 10 min, and washed with PBS three times. Cells were blocked by incubation with 10% BSA for 1 h at room temperature and subsequently washed with PBS three times. For antigen detection, samples were incubated with rabbit anti-collagen I antibody (Beijing Bioss, Beijing, China) at a dilution of 1:1,000 at 4˚C overnight and stained with a secondary FITC-labeled Goat anti-rabbit IgG antibody at a dilution of 1:1,000 for 1 h at room temperature (25˚C). Nuclei were

counterstained by using DAPI, and cells were subsequently observed by using a laser confocal microscope (LEICA; Sp-8).

## 2.9 Immunohistochemistry staining

Femoral tissue sections were dewaxed in xylene and rehydrated. Antigen retrieval was achieved by microwaving samples at 92°C for 10min. Sections were incubated with 0.3% hydrogen peroxide to inactivate endogenous peroxidase activity. After blocking with goat serum (2mg/ml), sections were incubated with primary antibodies (1: 100) against GLI2, IHH, SHH and RUNX2 (all Abcam, UK) at 4°C overnight. After washing with PBS, horseradish per-oxidase-conjugated secondary antibodies (1:5000) of IgG were added and incubated for 1h at room temperature (25°C). Sections were developed by using DAB chromogenic reagent, observed by using a light microscope, and then images were acquired. Grayscale values (OD) were calculated by using Image-Pro Plus 3.0 (IPP, USA).

## 2.10 BMD and Enzyme-linked immunosorbent assay (ELISA) determination

BMD was measured by dual energy X-ray absorptiometry by using the Lexxos type dual light energy X-ray bone density meter (DMS, France), computed by the Small Subject Scout Scan software, and expressed as bone mineral content per unit area (g/cm$^2$). Appropriate amount of femoral tissue was taken for homogenization, centrifuged at 2–8°C 5000 g for 5 min, and the supernatant was taken. Prior to examination, liquid supernatant was stored at– 20°C. The levels of SHH, IHH, Gli2, and Runx2 in bone tissue and levels of alkaline phosphatase (ALP), Bone Morphogenetic Protein (BMP), and type I collagen (COL1a1) in the culture supernatant were determined using a quantitative sandwich technique, for which ELISA kits (eBioscience, US) were used according to the manufacturer's instructions.

## 2.11 Quantitative real-time polymerase chain reaction (qRT-PCR)

The mRNA expression levels of *IHH*, *Shh*, *Gli2*, *DHH* and *Runx2* in BMSC cells after 7 days of compound incubation or femoral tissues were evaluated by using qRT-PCR. Total RNA was extracted with Trizol Reagent (TransGen Biotech, Beijing, China) and reverse transcribed by using the HiScript Reverse Transcriptase (RNase H) (Vazyme Biotech, Nanjing, China) to obtain cDNA. RT-qPCR was performed on an ABI7900 PCR (Illumina, CA, United states). The $2^{-\triangle\triangle CT}$ method was used for relative quantification of mRNA levels, with actin as the household gene control. The specificity of all PCR products was confirmed by melting curve analysis. Each experiment was conducted in triplicates. The primers used in this study are shown in S2 Table.

## 2.12 Western blotting

Cells were collected and subjected to cell lysis for 30 min at 4°C by using RIPA lysis buffer. The supernatant was collected by centrifuging samples at 12,000 rpm for 10 min. Total protein was extracted from femoral tissues and the protein concentration was determined by the BCA method. Isolated total protein was separated by using SDS-polyacrylamide gel electrophoresis and transferred to polyvinylidene fluoride membranes. After blocking membranes with 5% BSA at room temperature for 1 h, primary antibodies against β-actin, IHH, DHH, Shh, GLI2 (1:1000 dilution; Abcam, UK) were added and membranes were incubated at 4°C overnight. After washing with 1Xtbst three times, secondary antibodies (1: 5000) were added and membranes were incubated at 37°C for 2 h. Bands were visualized by using an enhanced

chemiluminescence kit (Pierce, Rockford, United States). Membranes were developed in a dark room and bands were analyzed by using the Image Quant software. The relative expression level was determined by the ratio of gray values of targets and β-actin bands, as evaluated by using Image Quant (GE Healthcare).

## 2.13 Bioinformatics analysis of Gene Expression Omnibus (GEO) database

To investigate genes associated with the pathogenesis of osteoporosis, we searched the GEO database to identify gene expression profiling studies of subjects with osteoporosis. We obtained gene expression profiles from GSE35958 which analyzed differential gene expression profiles between patients with OP and normal controls. All the data were freely available online. The GEO2R online analysis tool (https://www.ncbi.nlm.nih.gov/geo/geo2r/) was used to identify differentially expressed genes of OP patients, with FDR<0.05 and | Log2Foldchange|>1.0 being set as thresholds for differentially expressed genes [24, 25]. KEGG pathway enrichment analysis was performed by using the R package cluster profiler 3.0. A protein-protein interaction network was constructed via the STRING database and visualized by using cytoscape 3.0 [26]. All data were processed for normalization before differential analysis.

## 2.14 Statistical analysis

All statistical analyses were performed by using the SPSS package version 23.0 for Windows (SPSS, Chicago, IL, USA). Continuous variables were averaged, expressed as mean and standard deviation, and compared with analysis of covariance (ANOVA) and the Bonferroni's post hoc test. The null hypothesis was always rejected for values of $p < 0.05$. Repeated measurement design data was analyzed by using Repeated Measures Generalized Linear Models, which requires the spherical symmetry assumption to be satisfied. If the spherical symmetry was satisfied, the uncorrected F cutoff value was assumed; if the spherical symmetry assumption was not satisfied, the F-limit value corrected by Greenhouse-Geisser was used.

## 2.15 Network pharmacology analysis

A sample of 1g BSHX was collected and transferred to the EP tube. Then, 400μL extraction solution (acetonitrile: methanol = 1:1, internal standard mixture containing isotope labeling) was added, swirled for 30 s, ultrasounded in ice water bath for 10 min, and incubated for 1 h to precipitate protein. The sample was then centrifuged at 12,000r/min for 15 min. Transfer the supernatant to a fresh glass vial for analysis. LC-MS/MS analysis using UHPLC system (Vanquish, Thermo Fisher Scientific), UPLC BEH Amide column (2.1 mm × 100 mm, 1.7 mm) was used in conjunction with Q Exactive HFX mass spectrometry (Orbitrap MS, Thermo). The mobile phase consisted of 25 mmol/L ammonium acetate and 25 ammonia hydroxide, aqueous solution (pH = 9.75) (A) and acetonitrile (B), automatic injector temperature was 4˚C, and injection volume was 3 mL. The raw data is converted to mzXML format using ProteoWizard and processed using an internal program developed in R and based on XCMS for peak detection, extraction, alignment, and integration. The internal MS2 database was then applied for metabolite annotation. The cut-off value for the annotation is set to 0.3.

According to the metabolic information of BSHX, TCMSP database was used to analyze the bioactive compounds of BSHX. The corresponding targets of each metabolite was obtained, and the corresponding relationship between the metabolite and target was established. The protein name targets collected in TCMSP were converted into "gene symbol" format using STRING database, and all targets were combined and pooled together. Then, the targets that involved in OP pathogenesis was obtained using TTD, CTD, OMIM, PharmGKB and Drug-Bank databases. The Go and KEGG pathway enrichment analysis was performed using R

package Clusterprofiler 3.0. And the protein-protein interaction network was constructed on all targets using String database.

## 2.16 MTT assay

BSHX was ground into powder and extracted using 80% methyl alcohol. Then, the extracts were freeze-drying, and redissolved using dimethyl sulfoxide into 100mg/ml. Then, the 1g/ml BSHX extracts were diluted into different concentration (1000, 500, 250, 125, 60 μg/ml) using water for MTT assay. The cells (5x104 cells/ml) were seeded in 96-well plates (five wells per treatment group) and treated with various doses of BSHX extracts at 37°C. Cell viability was determined using methyl thiazol tetrazolium (MTT) (Sigma, USA) after a 48-hour incubation period by measuring optical density at a wavelength of 490 nm using a microplate reader (Denmark; BioRad).

## 3. Results

### 3.1 Hedgehog signaling-related genes involved in the pathogenesis of osteoporosis

To investigate the mechanisms of signaling-related genes in regulating the pathogenesis of osteoporosis, we analyzed transcription patterns of OP patients. Setting the FDR<0.05 and | Log2Foldchange|>1.0 as thresholds to identify differentially expressed genes (DEGs) in OP patients versus controls (Fig 1C), we found 3,194 DEGs, of which 2,542 were upregulated and 652 were downregulated (Fig 1A). Subsequently, KEGG pathway enrichment analysis on these 3,194 DEGs showed that these genes were involved in several key signaling pathways (Fig 1B), including regulation of actin cytoskeleton, the MAPK signaling pathway, signaling pathways regulating pluripotency of stem cells, the Chemokine signaling pathway, the Rap1 signaling pathway, the Wnt signaling pathway, and the Hedgehog signaling pathway. Various pathways above have been demonstrated to be involved in the differentiation of BMSCs, including Wnt, BMP/Smad, Notch, and Hedgehog [12, 14]. Then, we constructed a protein-protein interaction network of these signaling pathways, and noted that SHH and RUNX2 which involved in Hedgehog signaling pathway performed indispensable functions in the topological interaction network and had the highest weights in the network. Importantly, both IHH and RUNX2 were differentially expressed between OP patients and controls, suggesting an involvement of these genes in the pathogenesis of osteoporosis. Based on the weights of each gene in the network, we identified 30 genes as the hub genes (S3 Table). Further KEGG pathway enrichment analysis on these hub genes showed that the genes involved in the interaction network were involved in the Thyroid hormone signaling pathway, Hepatitis B, Basal cell carcinoma, TGF-beta signaling pathway, Th17 cell differentiation, Kaposi sarcoma-associated herpesvirus infection, Human T-cell leukemia virus 1 infection and, again, the Hedgehog signaling pathway (Fig 1D). We noted that various pathways relevant to cell differentiation were significantly enriched based on these genes, suggesting that Hedgehog signaling-related genes may be involved in the pathogenesis of osteoporosis and cell differentiation. Thus, genes involved in Hedgehog signaling pathway were important in pathogenesis of OP, especially *SHH*, *IHH*, *GLI2*, and *RUNX* that differentially expressed in OP patients.

### 3.2 BSHX increases the bone mineral density and osteogenic differentiation of rat BMSCs

Considering the relationship between Hedgehog signaling pathway and OP pathogenesis, we expected to investigate the involvement of related genes in the mechanisms of BSHX

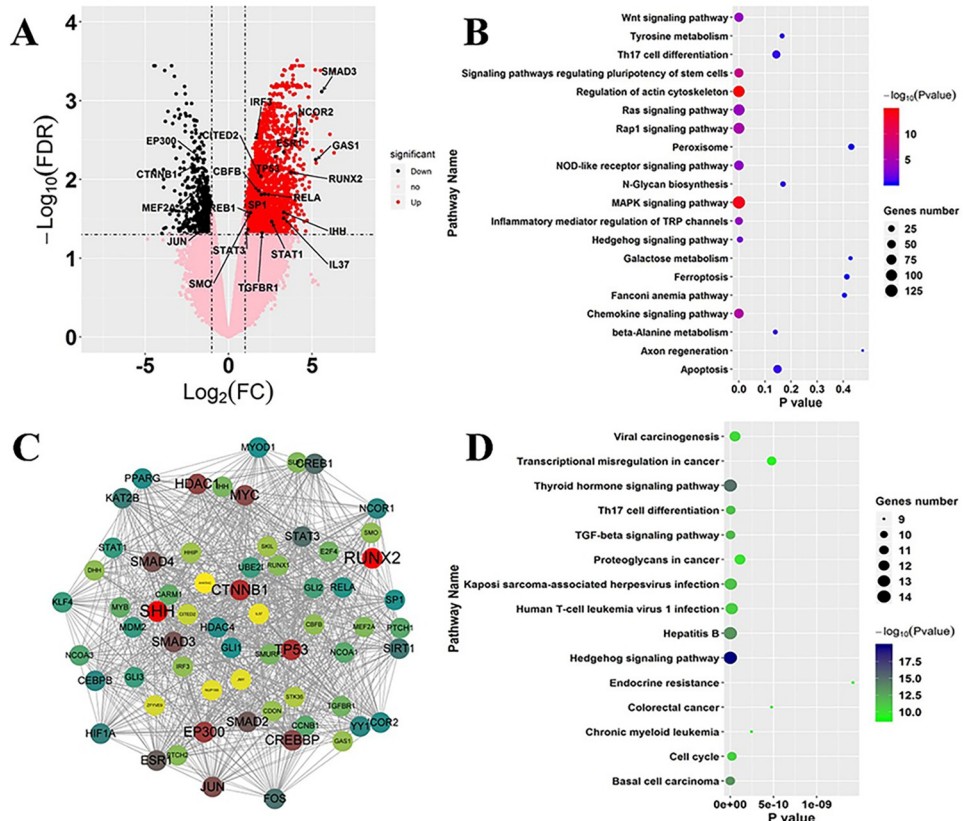

**Fig 1. Alterations in Hedgehog signaling-related genes are key characteristics in the transcriptomic landscape of OP patients. A:** Volcano plots showing the DEGs in OP vs. controls (Log2Foldchange>1.0; FDR<0.05). Upregulated and downregulated genes are shown in black and red, respectively, while pink dots represent genes without significant differences. The genes labeled by the black arrow are hub genes involved in the Hedgehog signaling pathway. **B:** Dot plot showing the results of KEGG pathway enrichment analysis on differentially expressed genes in OP vs. control comparisons. The dot color represents the P value of each pathway. **C:** Protein-protein interaction network of SHH, IHH, GLI2, and RUNX2. The dot color and label size represent the importance of each gene in this topological network. **D:** Dot plots showing KEGG pathway enrichment analysis results of genes involved in the protein-protein network.

improving OP. Thus, we established a rat model of OP to evaluate the molecular effects underlying the amelioration of OP using BSHX. MTT results showed that BSHX extracts did not affect the cell growth of mesenchymal stem cells (S2 Fig), implying that BSHX did not exert significant toxicity on cells. Fig 2A shows HE images of rats in the OP, BSHX, estradiol, and calcium groups. Compared with the blank control group, the structural integrity of the trabecular bone in the OP model and calcium group was observably incomplete, with some parts of the trabecular bone being damaged and broken (Fig 2A). Additionally, the reticular structure was destroyed and the bone marrow cavity was widened or irregular in OP model rats (Fig 2A). Remarkably, in the BSHX and estradiol (E2) groups, the structure of the trabecular bone was tightened, suggesting that BSHX treatment may improve the bone mineral density (Fig 2A). To confirm the positive effect of BSHX on BMD, we next quantified the BMD in each treatment group. The BMD in OP model rats was significantly lower than in controls ($0.15 \text{ g/m}^2$ versus $0.24 \text{ g/m}^2$) (Table 1), demonstrating that osteoporosis caused a decrease in bone mineral content. Twelve weeks of BHSX treatment following ovariectomy significantly increased the BMD ($0.22 \text{ g/m}^2$) compared to OP model rats, and there was no significant

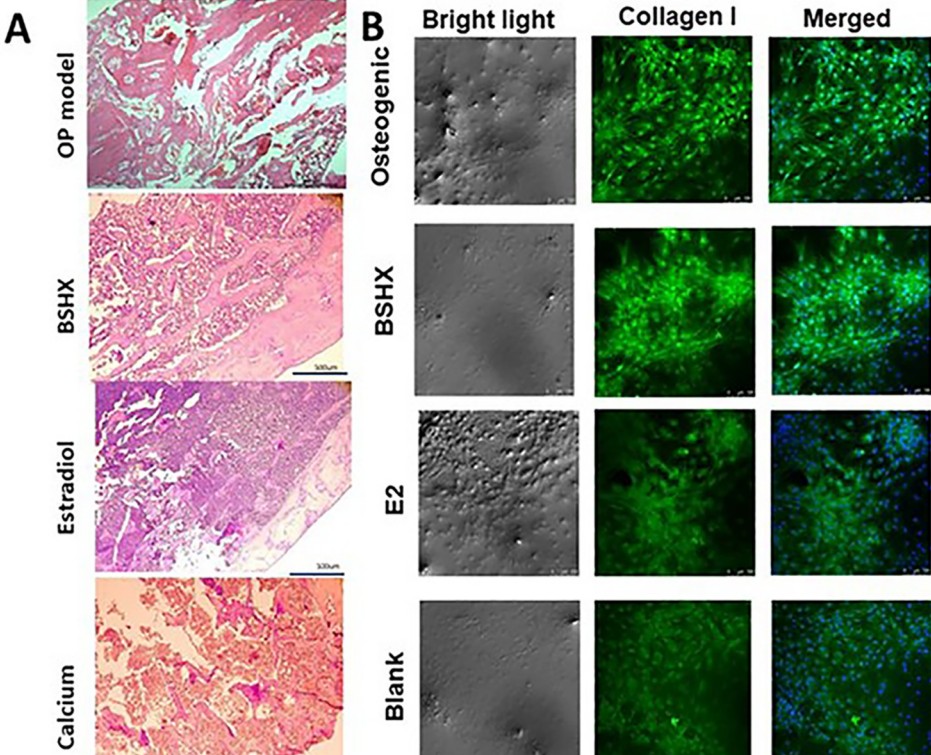

**Fig 2. Histological assessment of bone tissue in an osteoporosis rat model after BSHX treatment. A:** The morphology of femoral head in each experimental group under light microscope Scale bar, 500 μm. **B:** Immunofluorescence staining shows the expression of collagen I in BMSCs. The expression level of collagen I is represented by the degree of green fluorescence, while nuclei were counterstained by using DAPI (blue fluorescence). Scale bar, 100 μm.

difference between BSHX-treated OP model rats and controls ($P>0.05$; Table 1). Conversely, both estradiol and calcium treatment resulted in a significantly lower BMD than in controls (0.20 and 0.19 g/m$^2$, respectively) (Table 1; $P<0.05$), suggesting that BSHX had a more pronounced effect of restoring BMD than both E2 and calcium.

We next utilized collagen I immunofluorescence staining to explore the effect of BSHX on promoting osteogenesis of BMSCs, assessing intracellular expression of collagen I (Fig 2B). We found positive immunofluorescence for collagen I in both BSHX and osteogenic inducer treatments (Fig 2B), with the degree of intracellular collagen I immunofluorescence in blank and E2 groups being lower (Fig 2B). This suggested that both BSHX and osteogenic inducer

**Table 1. Comparison of the bone mineral density in five experimental groups.**

| Groups | Cases | BMD (g/m$^2$) |
|---|---|---|
| Blank | 10 | 0.239 ± 0.015 |
| OP model | 10 | 0.150 ± 0.045* |
| BSHX | 10 | 0.224 ± 0.032# |
| estradiol (E2) | 10 | 0.196 ± 0.038* |
| calcium | 10 | 0.190 ± 0.055* |

\* Compared with the blank group, $P<0.05$; # Between BSHX group vs. OP model group, $P<0.05$.

treatments induced the expression of collagen I in tissue. Taken together, the above results suggested that BSHX increased the BMD in our rat OP model and induced intracellular collagen I expression, which may promote the osteogenic differentiation of BMSCs.

### 3.3 Levels of SHH, IHH, GLI2, and RUNX2 in rat bone tissue

Our results indicated that BSHX treatment may promote BMSC differentiation (Fig 2). The Hedgehog signaling pathway has previously been shown to regulate BMSC differentiation [13], and therefore we hypothesized that BSHX may influence the Hedgehog signaling pathway. We therefore detected the optical density of Hedgehog signaling-related molecules SHH, IHH, GLI2, and RUNX2 by using immunohistochemistry (Fig 3A; Table 2). Animals in the OP model group exhibited the highest levels of SHH (162.1 ± 11.7), which were therefore almost 2-fold higher than in blank controls (Fig 3A; Table 2). SHH levels in rat tissues from calcium, BSHX, and E2 treatment groups were also higher than in controls (Fig 3A; Table 2). Moreover, levels of IHH were significantly decreased in all four treatment groups compared to controls (Fig 3A; Table 2). BSHX also increased levels of GLI2 in tissue, however there was no significant difference in RUNX2 levels between groups (Fig 3A; Table2).

We also conducted RT-qPCR to determine the effect of BSHX on mRNA expression levels of Shh, Ihh, Runx2, and Gli2. Ihh was not detected in bone tissue (Fig 3B). Runx2 and Gli2 expression levels were significantly higher in OP model rats than in the other groups, with the lowest expression levels of Runx2 in the blank control and Gli2 in calcium group (Fig 3B). BSHX did not affected mRNA expression of Runx2, but increased the Gli2 level, in line with immunohistochemistry results (Figs 2A; 3B). Conversely, BSHX treatment dramatically suppressed the mRNA expression of *Shh*, while calcium, E2 and OP did not significantly affect *Shh* expression (Fig 3B).

Lastly, we performed ELISA analysis to investigate protein abundance of SHH, IHH, GLI2, and RUNX2 (Fig 3C). Control animals exhibited highest expression of IHH protein, with levels significantly lower in OP group animals (Fig 3C). Of note, BSHX treatment significantly increased IHH protein levels (Fig 3C). Calcium treatment dramatically elevated SHH protein expression, while E2 significantly reduced it (Fig 3C). Although OP exhibited reduced protein levels compared with controls, there was no significant difference in SHH protein levels between the OP and BSHX groups (Fig 3C). GLI2 protein was highest in the BHSX group (Fig 3C). RUNX2 protein levels were highest in control animals and lowest in OP and calcium groups. Both E2 and BSHX increased RUNX2 protein levels (Fig 3C). Overall, these data indicated that BSHX may affect the Hedgehog signaling pathway by altering protein levels of IHH and GLI2 in bone tissue.

### 3.4 BSHX promotes the expression of ALP, BMP, and COL1A1 in BMSCs

Next, we tested concentrations of ALP, BMP, and COL1A1 1, 3, 5, and 7 days after BSHX induction. Using repeated measurement analyses of variance from a general linear model to analyze the time effect, group effect, and interaction between the two, we found a statistically significant time effect, group effect, and the interaction of ALP (F = 2457.3, $p < 0.001$; F = 54.78, $p < 0.001$; F = 606.48, $p < 0.001$; Fig 4A), BMP (F = 76.305, $p < 0.001$; F = 347.335, $p < 0.001$; F = 99.28, $p < 0.001$; Fig 4B), and COL1A1 (F = 342.58, $p < 0.001$; F = 113.25, $p < 0.01$; F = 38.45, $p < 0.001$; Fig 4C) between each intervention group and blank group. These results suggested that BSHX as well as estradiol and the osteogenic inducer promote the expression of COL1a1, BMP, and ALP in BMSCs with time and group effects.

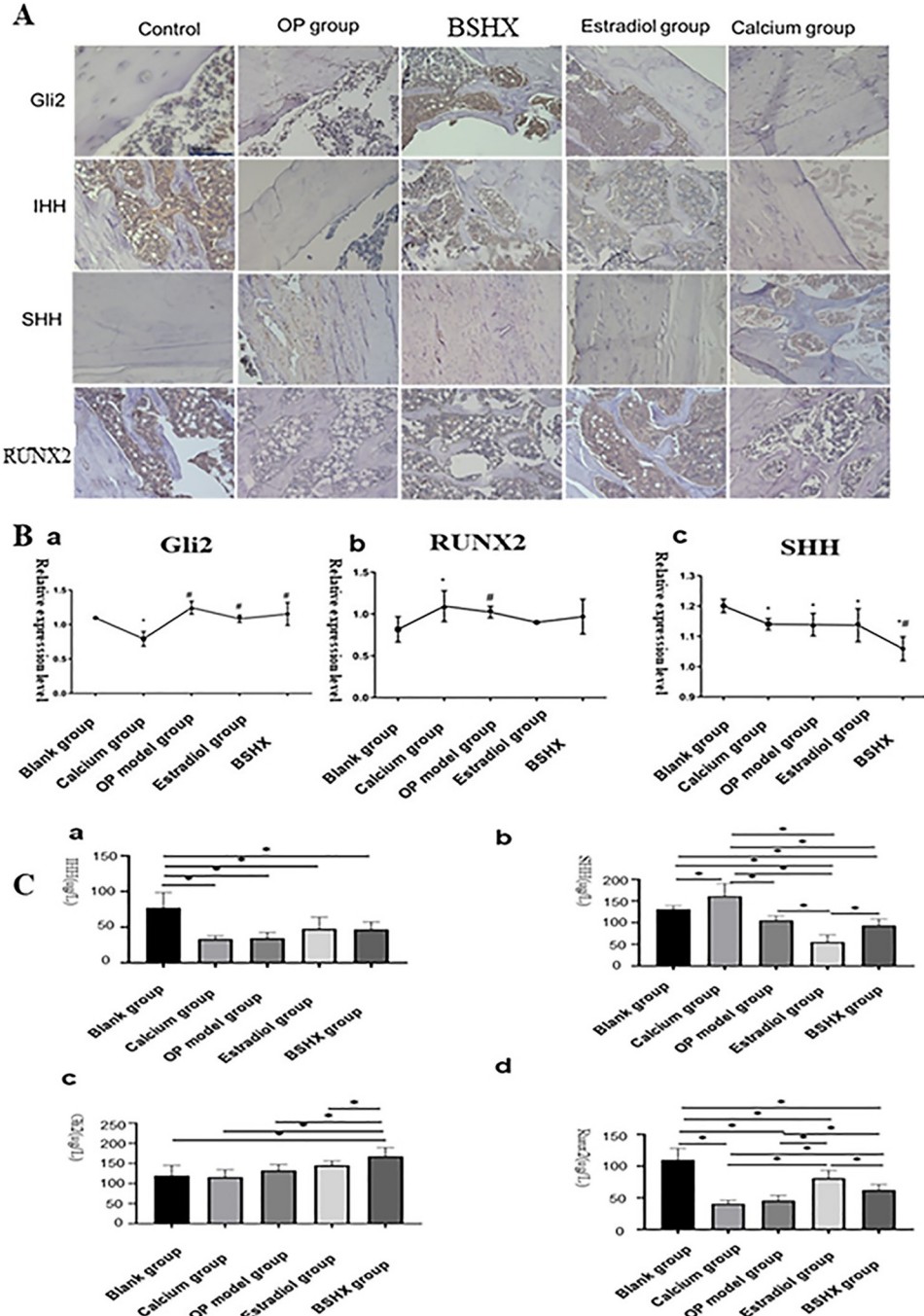

**Fig 3. Changes in the Hedgehog signaling pathway after BSHX treatment. A:** Immunohistochemical staining of SHH, IHH, GLI2, and RUNX2. Scale bar, 500 μm. **B:** RT-qPCR analysis of Shh, Ihh, Gli2, and Runx2 mRNA levels in rat bone tissue. *P<0.05 compared with Blank group; #P<0.05 compared with Calcium group. **C:** the concentrations of SHH, IHH, Gli2, and Runx2 in rat bone tissue were measured by ELISA. *, *P<0.05.*

## 3.5 BSHX upregulates *Shh*, *Ihh*, *Gli2*, and *Runx2* expression in BMSCs

Considering the effect of BSHX on SHH, IHH, GLI2, and RUNX2 protein levels in the bone of OP model rats, we further evaluated the expression of these molecules at the mRNA levels in

**Table 2. Comparison of the optical density of SHH, IHH, GLI2, and RUNX2 between groups.**

| Groups | Cases | IHH | SHH | Runx2 | Gli2 |
|---|---|---|---|---|---|
| blank | 10 | 170.01±11.417 | 85.71±13.145 | 147.78±13.576 | 131.24±11.860 |
| calcium | 10 | 135.69±9.580* | 144.36±13.530* | 136.64±12.090 | 140.35±12.811 |
| BSHX | 10 | 136.88±11.505* | 143.02±15.182* | 144.23±12.573 | 150.42±13.699* |
| E2 | 10 | 150.06±14.201* | 143.86±17.652* | 146.31±13.391 | 150.31±15.208 |
| OP model | 10 | 127.67±10.783* | 162.09±11.716* | 136.63±10.845 | 116.91±12.857 |

*, P<0.05

**, P<0.01.

BMSCs after BSHX treatment by using RT-qPCR. Expression levels of SHH, GLI2 and RUNX2 genes were lowest in blank controls, while osteogenic inducer, BSHX, and E2 treatments increased the expression levels of all these genes to varying degrees (*P*<0.05; Fig 5A): osteogenic inducer dramatically increased the expression of all four genes (Fig 5A). BSHX also increased their expression to higher levels than in controls (Fig 5A). Conversely, the effect of E2 on inducing *Shh*, *Ihh*, *Gli2*, and *Runx2* expression in BMSCs was weaker than that of BSHX and osteogenic inducer (Fig 5A). While the induction effect of BSHX on gene expressions was relatively weaker than that of osteogenic inducer treatment, these results demonstrated that BSHX treatment still effectively increased Shh, Ihh, Gli2, and Runx2 expression levels (Fig 5A). To further explore the relationship between BSHX and the Hedgehog pathway, we next used Western blotting to evaluate protein levels of SHH, IHH, GLI2 and RUNX2 (Fig 5B). Both osteogenic inducer and BSHX treatments significantly increased protein levels of GLI2 and SHH compared to other groups, while they were relatively higher in BMSCs from osteogenic inducer and control groups (*P*<0.05; Fig 5B). BSHX also reduced the IHH level in BMSCs and did not affect RUNX2 (Fig 5B). Additionally, we also detected the expression level of *DHH* in OP model before and after BSHX treatment. The results showed that *DHH* was upregulated in OP model, and its expression level was suppressed to downregulate in OP model after BSHX and E2 treatments (S1B Fig). Consistently, western blot results showed that the level of DHH protein was higher in OP model compared to control (blank), while BSHX treatment significantly reduced the levels of DHH in OP model rats (S1A Fig). Overall, these results suggested that BSHX induced the expression of hub genes involved in the Hedgehog signaling pathway, potentially thereby promoting the differentiation of BMSCs.

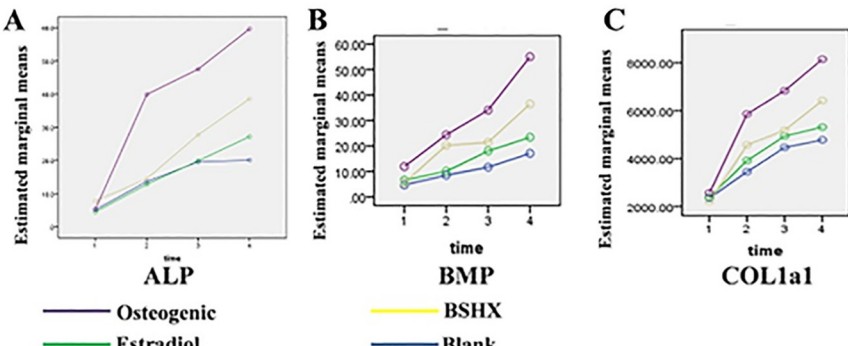

**Fig 4. Effect of BSHX on expression levels of ALP, BMP, and COL1A1 in BMSCs.** The concentration of ALP (**A**), BMP (**B**) and COL1A1 (**C**) in BMSCs after BSHX, osteogenic inducer, or estradiol treatment.

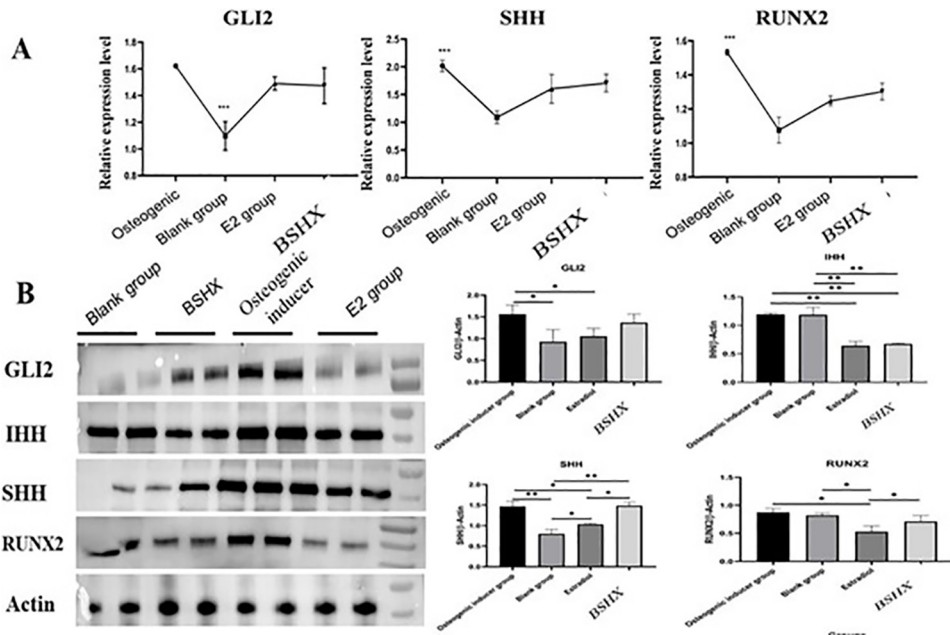

**Fig 5. mRNA and protein expression levels of SHH, IHH, GLI2, and RUNX2 in BMSCs after BFHX treatment. A:** qRT-PCR results displayed the mRNA expression levels of *Shh*, *Gli2*, and *Runx2* in BMSCs after BFHX treatment. **B:** Western blot and quantification of protein levels of SHH, IHH, GLI2 and RUNX2 in BMSCs after BSHX treatment. *, $P<0.05$; **, $P<0.01$.

## 3.6 Quercetin is the hub active compounds of BSHX in improving OP via affecting Hedgehog signaling pathway

To investigate the bioactive compounds of BSHX, we constructed the metabolic fingerprint of BSHX using LC-MS/MS platform. Totally, 503 metabolites were identified in BSHX extracts, and these metabolites were mainly belonged to Amino acids, Saturated Fatty Acids, C30 iso-prenoids, Hydroxycinnamic acids, C10 isoprenoids, Flavones, Alkanes, Hydrocarbons, Stig-masterols, Isoflavonoids, Diterpenoids, Tyrosols, Fatty alcohols, Salicylic acids, Gallic acids, Hydroxyflavonoids, Aryl-aldehydes, Isoflavonoid O-glycosides, Cholines, M-methoxybenzoic acids and Coumarins (Fig 6A; S4 Table). Then, we performed Network pharmacological analysis on these 503 metabolites to identify their candidate targets using TCMSP database (https://tcmsp-e.com/index.php) (S5 Table). The results showed that there were 1035 proteins were identified as the targets of these metabolites (S5 Table). Then, we performed GO and KEGG pathway enrichment analysis on these targets to investigate their functions in vivo. The results of GO enrichment showed that these targets mainly performed their functions in Extracellular space, Cytoplasm, Cell surface, Vesicle, Extracellular exosome, Cellular anatomical entity, Secretory granule, Mitochondrion, Vesicle lumen and Plasma membrane region, and exerted numerous activities, including Identical protein binding, Protein binding, Catalytic activity, Enzyme binding, Ion binding, Small molecule binding, Anion binding, Oxidoreductase activity, Vitamin binding and Signaling receptor binding (S6 Table). And these proteins involved in numerous important biological progresses, including Response to oxygen-containing compound, Response to organic substance, Regulation of biological quality, Response to organic cyclic compound, Response to stimulus, Response to nitrogen compound, Response to stress, Response to lipid, small molecule metabolic process, Organic substance metabolic process, Positive regulation of biological process and Metabolic process (S6 Table). In parallel, KEGG

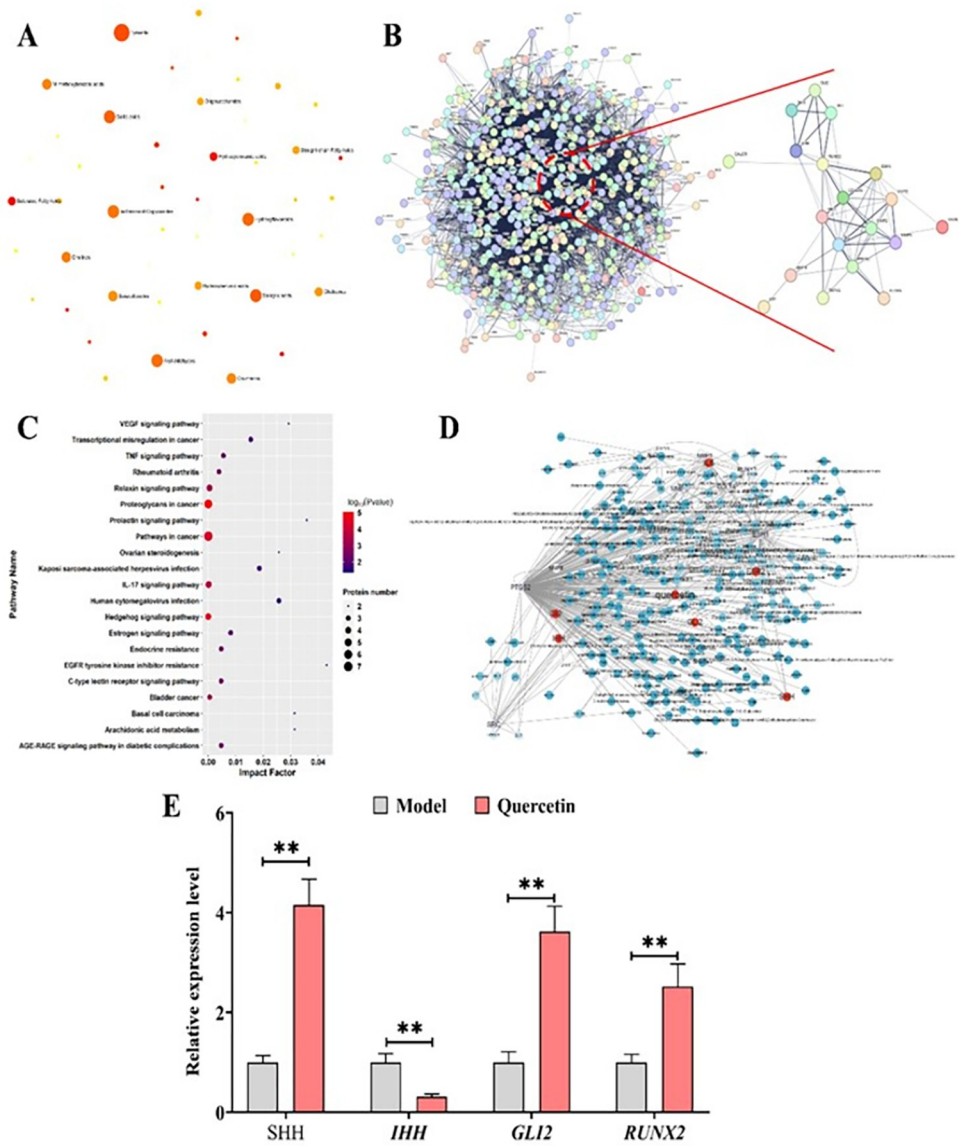

**Fig 6. Quercetin is the hub active compounds of BSHX in improving OP. A:** Enrichment analysis of metabolites in BSHX. **B:** Protein-protein interaction network of targets of metabolites in BSHX. **C:** Scatter plot showing the results of KEGG pathway enrichment analysis on targets of bioactive compounds in BSHX. The dot color represents the P value of each pathway. **D:** Pharmacology network of targets of bioactive compounds in BSHX. **E:** RT-qPCR assay detects the expression levels of Hedgehog-related genes in OP model rats after quercetin treatment.

pathway analysis on these targets showed that these proteins mainly involved in 245 pathways, especially Pathways in cancer, Metabolic pathways, PI3K-Akt signaling pathway, Prostate cancer, HIF-1 signaling pathway, Neuroactive ligand-receptor interaction, MicroRNAs in cancer, TNF signaling pathway, Hedgehog signaling pathway and Osteoclast differentiation (S7 Table). We noted the significant enrichment of Hedgehog signaling pathway and Osteoclast differentiation, implying the involvement of both pathways in the mechanisms of BSHX-improving OP. Then, we constructed the protein-protein interaction network of these targets using string database (Fig 6B). The results showed that these targets of BSHX formed a complex interaction network, implying that BSHX might improve OP via a complex strategy

(Fig 6B). Of note, we identified a hub interaction node in this network, which was constructed by 18 proteins that involved in Hedgehog pathway and OP pathogenesis (Fig 6B). KEGG enrichment analysis on these 18 proteins showed that these proteins mainly functioned in Hedgehog signaling pathway, Osteoclast differentiation, Pathways in cancer, IL-17 signaling pathway and TNF signaling pathway (Fig 6C), supporting that BSHX could improve OP via targeting Hedgehog signaling pathway. Typically, we identified that 18 proteins that involved in OP pathogenesis were the targets of the compounds of BSHX, including IL6, MMP1, ESR1, MMP3, ALOX5, COL1A1, PTGS2, MMP2, FASN, LTF, TRPV1, MMP8, CALCR, SHH, SRC, IHH, Gli2 and Runx2 (Fig 6D). Typically, we found that quercetin could target ALOX5, COL1A1, IL6, MMP1, MMP2, MMP3, PTGS2, RUNX2, SHH, IHH and Gli2 (Fig 6D). Then, we performed RT-qPCR assay to investigate the effects of quercetin in affecting these genes in OP model. The results showed that quercetin treatment exhibited similar effects as BSHX that quercetin treatment (50μg/ml) significantly induced the expression of *SHH*, *Gli2* and *Runx2*, and inhibited the expression of *IHH* in OP model rats (Fig 6E). Thus, these results suggested that quercetin was the active compounds of BSHX that involved in the effects of BSHX in improving OP.

## 4. Discussion

Abnormal osteogenic differentiation of BMSCs is involved in the pathogenesis of osteoporosis. Thus, BMSCs are considered as a potential target for medical interventions the treatment of osteoporosis [27]. Our previous studies demonstrated that BSHX promotes the osteogenic differentiation of BMSCs [6]. Presently, our study observed that BSHX treatment effectively increased the BMD content and improved the microstructure of bones in an OP rat model. Immunohistochemistry indicated that BSHX improves the BMD in OP rats by inhibiting the destruction of the trabecular bone and reticular structure. The pathogenesis of OP has previously been shown to be closely associated with alterations in the expression of genes involved in the Hedgehog signaling pathways, including Shh, Ihh, Gli2, and Runx2, which are also involved in cell differentiation. IHH regulates osteogenesis by promoting osteoblast differentiation of mesenchymal cells. IHH exert its biological effects through its receptor components patched (PTCH). Gli2 function as direct mediators for IHH signaling. Consistently, as hedgehog family member, SHH2 acts synergistically with IHH to stimulate osteoblast differentiation of mesenchymal cells [28]. Shimoyama A et al. have found that Gli2 up-regulated Runx2 expression and enhanced the osteogenic action of Runx2 through physical association with Runx2 [29]. Although effect of BSHX on expression of these genes varied at mRNA and protein levels, BSHX effectively triggered the expression levels of hub Hedgehog signaling-related molecules, especially IHH and GLI2 in both bone tissue and BMSCs of osteoporotic rats. The varied expression levels of SHH and RUNX2 among bone tissue and BMSCs under BSHX treatment may cause by different levels of related genes in different tissue. Importantly, our network pharmacology results identified that quercetin was the active compounds that involved in the mechanism of BSHX-improved OP via affecting Hedgehog-related genes. We conclude that BSHX may redirect the differentiation of BSMCs to osteoblasts by activating the Hedgehog signaling pathway, ultimately achieving effective improvements in osteoporosis.

The pathological manifestations of OP are trabecular bone thinning and microfractures that cause intra-osseous hemorrhage [50]. In the bone marrow of OP patients, the balance of BMSC differentiation is disrupted which results in decreased osteogenic cell production [30–32]. Unfortunately, most therapeutics used to date have severe adverse effects. High-dose and long-term use of teriparatide, a parathyroid hormone analog, promotes bone formation but increases the incidence of osteosarcoma in rats [33]. Strontium ranelate has the dual effects of

inhibiting bone resorption and promoting bone formation, but also has the risk of inducing cardiovascular and cerebrovascular diseases [34]. TCM have grown in popularity as a result of its success in curing ailments while causing minimal adverse effects [35]. Recent scientific reports suggest that some classic and bone-specific natural Chinese medicine are very popularly used to treat osteoporosis and bone fracture effectively in clinical with their potential value in bone growth and development, but with few adverse side-effects. Some TCMs such as antler, epimedium, psoralen and salvia were the most classical and bone-specific of drugs when applied to the treatment of bone loss diseases, with the effects on the growth and development of skeleton tissue [36]. We observed that BSHX effectively improved trabecular bone and reticular structure, which is consistent with previous [36, 37]. In this study, we further demonstrated that BSHX promotes the osteogenic differentiation of BMSCs. As the main functional cells for bone formation, osteoblasts express specific markers such as ALP during differentiation and maturation, and ALP, COL1A1 and BMP, which are generally used as specific indicators for osteoblast identification [38, 39]. BSHX increased the concentrations of ALP, BMP, and COL1A1 in BMSCs, suggesting that BSHX promotes BMSCs differentiation.

The bone metabolism includes osteoblast differentiation, formation, absorption, and destruction. This process is complex and regulated by multiple signaling pathways, prominent the Hedgehog pathway [21]. Hedgehog-related signaling is triggered to perform functions in the early stages of osteoblast differentiation, and Hedgehog signaling gradually declines with the maturation of osteoblasts [40]. Results from the present study indicate that Hedgehog-related signaling molecules IHH and GLI2 were increased by BSHX treatment, suggesting an involvement of the Hedgehog pathway in mediating BSHX-induced effects of osteogenesis. IHH and GLI2, the main signaling molecules of the Hedgehog pathway, have distinct roles in osteogenesis [41, 42]. The GLI protein family is a nuclear transcription factor. GLI1 and GLI2 are transcription-promoting factors, while GLI3 is a transcriptional repressor [43, 44]. RUNX2 plays an important role in osteogenic differentiation and bone formation and is a key transcription factor initiating osteogenesis [45]. It regulates the expression of major bone-forming proteins such as osteocalcin, osteopontin, and bone sialoprotein, [46–48]. Previous studies have shown that GLI2 upregulates RUNX2 expression, which is regulated by various signals such as bone morphogenetic proteins [29, 49–51]. Ozturk et al. found that treatment with resveratrol increased the expression of *ALP* and *Runx2*, which have a protective effect on bone strength and general microarchitectural properties [51]. In the present study, BSHX could reduce osteoporosis symptoms in bone through RUNX2 upregulation. SHH is one of the key signals in the early stages of osteoblast differentiation, while IHH is involved in regulating late differentiation [41, 42, 52], consistent with our findings that BSHX improve OP through promoting the expression of COL1a1, BMP, and ALP in BMSCs with time and group effects. And the upregulation of SHH in BSMCs under BSHX treatment supported the key role of SHH in osteoblast differentiation. However, the detailed functions of SHH and IHH in osteogenic induction of BMSCs still require further in-depth research.

Here, we demonstrate that BSHX improved osteoporosis by increasing the BMD and inhibiting the destruction of trabecular bone and reticular structure in rats. BSHX promoted the osteogenic differentiation of BMSCs by modifying IHH and GLI2, key molecules related to the Hedgehog pathway. Of note, quercetin was the bioactive compounds of BSHX in improving OP. Differences in protein and mRNA expression we observed may be caused by different sources of cellular material, stages of differentiation, miRNA regulation, and kinase activation. Therefore, the specific mechanism of action and the overall pharmacodynamics and pharmacokinetics of BSHX will need to be further studied. Nonetheless, our present results provide valuable initial insights into the molecular mechanisms that BSHX improves osteoporosis, forming the basis for further BSHX-based osteoporosis treatments.

## 5. Conclusion

BSHX can effectively improve BMSC differentiation into osteoblasts by increasing the expression of Hedgehog signaling-related genes *Shh*, *Ihh*, *Gli2, and Runx2*. And quercetin was the main bioactive compounds of BSHX in improving OP via affecting Hedgehog signaling pathway.

## Supporting information

**S1 Fig. DHH of bone tissue in an osteoporosis rat model after BSHX treatment.**
(TIF)

**S2 Fig. MTT results of mesenchymal stem cells treated with BSHX extract.**
(TIF)

**S1 Table. Composition of BSHXF.**
(XLSX)

**S2 Table. Primers in the experiment.**
(XLSX)

**S3 Table. Characteristics of proteins invovlved in the network.**
(XLS)

**S4 Table. Metabolic compositions in of BSHXF.**
(XLSX)

**S5 Table. Candidate targets of the compounds in BSHXF from TCMSP database.**
(XLSX)

**S6 Table. Go enrichment results on the candidate targets of BSHXF.**
(XLSX)

**S7 Table. KEGG pathway enrichment results on the candidate targets of BSHXF.**
(XLSX)

## Acknowledgments

The authors would like to thank all the reviewers who participated in the review, as well as MJ Editor (www.mjeditor.com) for providing English editing services during the preparation of this manuscript.

## Author Contributions

**Conceptualization:** Yuqi Chen.

**Data curation:** Yuqi Chen, ZhiYong Wei, HongXia Shi, Xin Wen, YiRan Wang, Rong Wei.

**Formal analysis:** Yuqi Chen.

**Funding acquisition:** Rong Wei.

**Investigation:** Yuqi Chen, Rong Wei.

**Methodology:** Yuqi Chen, ZhiYong Wei, HongXia Shi, Xin Wen, YiRan Wang.

**Project administration:** Yuqi Chen.

**Resources:** Rong Wei.

**Supervision:** Rong Wei.

**Validation:** Rong Wei.

**Writing – original draft:** Yuqi Chen, ZhiYong Wei, HongXia Shi, Xin Wen, YiRan Wang, Rong Wei.

**Writing – review & editing:** Yuqi Chen, ZhiYong Wei, Rong Wei.

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
