## [Decision Letter · Decision Letter 0]

7 Jun 2023

PONE-D-23-01680BushenHuoxue Formula improves osteoporosis by promoting Osteogenic Differentiation of Bone Marrow Stem CellsPLOS ONE

Dear Dr. Wei,

Thank you for submitting your manuscript to PLOS ONE. After careful consideration, we feel that it has merit but does not fully meet PLOS ONE’s publication criteria as it currently stands. Therefore, we invite you to submit a revised version of the manuscript that addresses the points raised during the review process.

We look forward to receiving your revised manuscript.

Kind regards,

Gary S. Stein

Academic Editor

PLOS ONE

Journal Requirements:

3. PLOS requires an ORCID iD for the corresponding author in Editorial Manager on papers submitted after December 6th, 2016. Please ensure that you have an ORCID iD and that it is validated in Editorial Manager. To do this, go to ‘Update my Information’ (in the upper left-hand corner of the main menu), and click on the Fetch/Validate link next to the ORCID field. This will take you to the ORCID site and allow you to create a new iD or authenticate a pre-existing iD in Editorial Manager. Please see the following video for instructions on linking an ORCID iD to your Editorial Manager account: https://www.youtube.com/watch?v=_xcclfuvtxQ.

Reviewers' comments:

Reviewer's Responses to Questions

**Comments to the Author**

1. Is the manuscript technically sound, and do the data support the conclusions?

Reviewer #1: Partly

Reviewer #2: Yes

2. Has the statistical analysis been performed appropriately and rigorously? 

Reviewer #1: Yes

Reviewer #2: Yes

3. Have the authors made all data underlying the findings in their manuscript fully available?

Reviewer #1: Yes

Reviewer #2: Yes

4. Is the manuscript presented in an intelligible fashion and written in standard English?

Reviewer #1: No

Reviewer #2: Yes

5. Review Comments to the Author

Reviewer #1: Comments to authors:

This manuscript reported an interesting topic that the Chinese traditional medicine BSHX promotes osteoporosis. However, there are multiple concerns to be addressed.

Specific comments:

1. The supplementary materials were missed.

2. The title may need to be formatted.

3. Grammar and some expressions need to be corrected and simplified.

4. Mammals have three Hedgehog homologues, you only mentioned IHH and SHH, how about DHH in Hedgehog signaling pathway?

5. The description of BSHX was missed. What components are in BSHX? Which are effective?

6. Line 132 and 133, the first “after” should be removed and what is “every t3 days”?

7. Line 141, correct the typo “10-8 mol/L”.

8. Reformat line 168-172.

9. Lack of evidence that supports the relationship between SHH, IHH, GLI2, RUNX2 and pathogenesis of OP.

10. With OD, qRT-PCR, ELISA analysis, you concluded that BSHX may affect the Hedgehog signaling pathway by altering protein levels of IHH and GLI2 in bone tissue, but how BSHX alter IHH and GLI2? And how about SHH and RUNX2?

11. Side effects and toxicity of BSHX are missed. This part is important.

Reviewer #2: Introduction

There was no aim of the stıdy and hypothesis. Please add the end of the introduction section.

Surgery and sacrificing procedures should be explained in more detail.

Material and methods

“Briefly, 10 kg of antler, epimedium, psoralen, salvia, astragalus, radix rehmanniae, white peony root, and other Chinese medicines were mixed with 1,500 mL boiling (100°C) distilled water and incubated by shaking at room temperature for 1 h”. This subject is not clear. How many kilograms of plants were used? Was it a total of 10 kg or 10 kg of each used? Is this mixture designed for this study or is it a previously used mixture for treatment? Add references, please.

İn 126. What does mean 10 female rats? Which group?

Please add references in section 2.5

Results

Figures 3 and 5: It is too hard to understand which result is significant or which is not. Figure 3 and 5 need to be redesigned.

İn 348. Why did you compare others to the calcium group? İt should be explained

Discussion

Runx 2 is a very important marker in bone metabolism and osteoporosis. Runx 2 needs to be more discussed in the discussion section. The following article should be used in this section

J Bone Miner Metab 2023 Apr 8.

doi: 10.1007/s00774-023-01416-z.

Resveratrol prevents ovariectomy-induced bone quality deterioration by improving the microarchitectural and biophysicochemical properties of bone

İn 420 and 421. “We observed that BSHX effectively 421 improved trabecular bone and reticular structure without any observable side effects.” How did you decide that BSHX does not effect any other tissue?

In addition to these, the following articles should be used in the discussion section.

Role of Traditional Chinese Medicine in Bone Regeneration and Osteoporosis

Zhicai Peng, Ronghua Xu, Qinjian You

Front Bioeng Biotechnol. 2022; 10: 911326

Identification of gene biomarkers in patients with postmenopausal osteoporosis

Chenggang Yang, Jing Ren, Bangling Li, Chuandi Jin, Cui Ma, Cheng Cheng, Yaolan Sun, Xiaofeng Shi

Mol Med Rep. 2019 Feb; 19(2): 1065–1073

Therapeutic Anabolic and Anticatabolic Benefits of Natural Chinese Medicines for the Treatment of Osteoporosis

Jianbo He, Xiaojuan Li, Ziyi Wang, Samuel Bennett, Kai Chen, Zhifeng Xiao, Jiheng Zhan, Shudong Chen, Yu Hou, Junhao Chen, Shaofang Wang, Jiake Xu, Dingkun Lin

Front Pharmacol. 2019; 10: 1344

There are a large number of grammatical errors in the manuscript. These errors need to be fixed

6. PLOS authors have the option to publish the peer review history of their article (what does this mean?). If published, this will include your full peer review and any attached files.

Reviewer #1: No

Reviewer #2: No

---

## [Author Response · Author response to Decision Letter 0]

18 Jul 2023

Dear Editors and Reviewers:

Thank you for your letter and for the reviewers’ comments about our manuscript entitled “BushenHuoxue Formula improves osteoporosis by promoting Osteogenic Differentiation of Bone Marrow Stem Cells” (PONE-D-23-01680). Those comments are all valuable and very helpful for revising and improving our paper, as well as the important guiding significance to our research. We have studied comments carefully and have made correction which we hope meeting with approval. Revised portions are highlighted in the paper. The main corrections in the paper and the responds to the reviewer's comments are as flowing:

Responds to the reviewer's comments:

Reviewer #1:

1. The supplementary materials were missed.

Response: Thanks for your revisions. We have submitted all the supplementary materials involved in the article, and some new experimental results as you suggested were added as supplementary materials.

2. The title may need to be formatted.

Response: Yes, thanks for your comments. And the title was revised as “BushenHuoxue formula promotes osteogenic differentiation via affecting Hedgehog signaling pathway in bone marrow stem cells to improve osteoporosis symptoms.”

3. Grammar and some expressions need to be corrected and simplified.

Response: Thanks for your comments. We have corrected the related errors of grammar and expression.

4. Mammals have three Hedgehog homologues, you only mentioned IHH and SHH, how about DHH in Hedgehog signaling pathway?

Response: Good suggestion. As you suggested, we performed related experiments to determine the expression of DHH in our experimental conditions, and the results are shown in the figure S1. Indeed, previous studies have suggested that the key factors of hedgehog signaling pathway involved in osteoporosis are SHH, GLI and IHH. And DHH only acts on gonads, including sperm and ovarian granulosa cells. Therefore, we did not detect DHH in the initial experiment. As you suggested, we found that DHH may also involve in mechanisms of BSHX and osteoporosis.

5. The description of BSHX was missed. What components are in BSHX? Which are effective?

Response: Thanks. We have added on the description of BSHX in “Introduction” and the components of BSHX in tableS1. Moreover, we further performed network pharmacology analysis to identify the main bioactive metabolites of BSHX in improving osteoporosis. And the related results were added in Fig. 6, we found that quercetin might be the main bioactive compounds in BSHX. Numerous proteins that involved in osteoporosis and hedgehog signaling pathway were the candidate targets of quercetin, including SHH, IHH, Gli2, etc. 

6. Line 132 and 133, the first “after” should be removed and what is “every t3 days”?

Response: Thank you for your suggestion. We have modified the mistakes.

7. Line 141, correct the typo “10-8 mol/L”.

Response: Thanks. We have corrected it.

8. Reformat line 168-172.

Response: Thanks for your suggestion. We have redescribed the expression.

9. Lack of evidence that supports the relationship between SHH, IHH, GLI2, RUNX2 and pathogenesis of OP.

Response: Thanks for your suggestion. We have added the evidence and references that support the relationship between OP and SHH, IHH, GLI2, RUNX2. The contents were shown in lines538-44 of the manuscript.

10. With OD, qRT-PCR, ELISA analysis, you concluded that BSHX may affect the Hedgehog signaling pathway by altering protein levels of IHH and GLI2 in bone tissue, but how BSHX alter IHH and GLI2? And how about SHH and RUNX2?

Response: Thanks for your reminding. To clarify how BSHX alter IHH and GLI2, we constructed the metabolic fingerprint of BSHX using LC-MS/MS platform. We have found that BSHX mainly perform its function in improving OP through quercetin. Quercetin could target and bind with several functional proteins that involved in OP and Hedgehog signaling pathway. RT-qPCR assay detects that quercetin could affected the expression levels of hedgehog-related genes, including IHH, Gli2, SHH and Runx2, then leading changes in Hedgehog signaling pathway, and OP pathogenesis. Remarkably, IHH and GLI2 were also identified as the targets of quercetin. Therefore, we considered that quercetin is the bioactive compounds of BSHX in improving OP via affecting Hedgehog signaling pathway.

11. Side effects and toxicity of BSHX are missed. This part is important.

Response: Yes, thanks for your reminding, we have carried out the MTT assay. MTT results showed that BSHX extracts did not affect the cell growth of mesenchymal stem cells (Figure S2). Thus, we proposed that BSHX might exert small side-effect in affecting human health or cell growth.

Reviewer #2:

1. There was no aim of the study and hypothesis. Please add the end of the introduction section. The surgery and sacrificing procedures should be explained in more detail.

Response: Thanks for your suggestion. We have added the aim of the study and hypothesis. Furthermore, the surgery and sacrificing procedures have been supplemented in the manuscript.

2. “Briefly, 10 kg of antler, epimedium, psoralen, salvia, astragalus, radix rehmanniae, white peony root, and other Chinese medicines were mixed with 1,500 mL boiling (100°C) distilled water and incubated by shaking at room temperature for 1 h”. This subject is not clear. How many kilograms of plants were used? Was it a total of 10 kg or 10 kg of each used? Is this mixture designed for this study or is it a previously used mixture for treatment? Add references, please.

Response: Thanks for your reminding, we have added the dose for each plant. And we have added the reference to show that BSHX was a previously used mixture.

3. İn 126. What does mean 10 female rats? Which group?

Response: Thank you for your suggestion. We have revised the expression. What 's more, this part of the rats was used to obtain BMSCs from SD rats at 6 weeks of age and were not grouped.

4. Please add references in section 2.5

Response: Thanks. We have added the reference in section 2.5.

5. Figures 3 and 5: It is too hard to understand which result is significant or which is not. Figures 3 and 5 need to be redesigned.

Response: Thanks for your suggestion. The Figure 3 mainly verifies the expression of SHH, IHH, Gli2, and Runx2 in bone tissue. The Figure 5 mainly describes the expression of factors in BMSCs. Our aim was to verify in vitro and in vivo that BSHX can improve osteoporosis by influencing the expression of some cytokines in the Hedgehog signaling pathway.

6. İn 348. Why did you compare others to the calcium group? İt should be explained.

Response: Thank you for your suggestion. Calcium supplement is the most basic treatment for osteoporosis. In this study, we compared other groups with calcium supplement group to investigate the effects of these treatment in improving OP symptoms.

7. Runx2 is a very important marker in bone metabolism and osteoporosis. Runx2 needs to be more discussed in the discussion section.

Response: Thanks for your reminding, we have added more references to discuss the function of Runx2 in OP pathogenesis and BSHX-treatment in the discussion section, and the references that you suggested have been cited in the new manuscript.

8. İn 420 and 421. “We observed that BSHX effectively 421 improved trabecular bone and reticular structure without any observable side effects.” How did you decide that BSHX does not effect any other tissue?

Response: Thanks for your suggestion. We have modified the inappropriate expression and made a more appropriate description.

9. In addition to these, the following articles should be used in the discussion section.

Response: This is a good proposal. We've read the corresponding literatures and cited them.

10. There are a large number of grammatical errors in the manuscript. These errors need to be fixed.

Response: Thanks for your comments. As your comments, I have recorrected some errors in grammar and expression.

---

## [Editor Report · Decision Letter 1]

31 Jul 2023

BushenHuoxue formula promotes osteogenic differentiation via affecting Hedgehog signaling pathway in bone marrow stem cells to improve osteoporosis symptoms

PONE-D-23-01680R1

Dear Dr. Wei,

We’re pleased to inform you that your manuscript has been judged scientifically suitable for publication and will be formally accepted for publication once it meets all outstanding technical requirements.

Kind regards,

Gary S. Stein

Academic Editor

PLOS ONE
---

## [Editor Report · Acceptance letter]

3 Aug 2023

PONE-D-23-01680R1 

BushenHuoxue formula promotes osteogenic differentiation via affecting Hedgehog signaling pathway in bone marrow stem cells to improve osteoporosis symptoms. 

Dear Dr. Wei:

I'm pleased to inform you that your manuscript has been deemed suitable for publication in PLOS ONE. Congratulations! Your manuscript is now with our production department. 

Kind regards, 

on behalf of

Dr. Gary S. Stein 

Academic Editor

PLOS ONE